# Knowledge, Attitudes, Perceptions, and Acceptance of COVID-19 Vaccination among Pharmacy and Non-Pharmacy Students

**DOI:** 10.3390/vaccines11010176

**Published:** 2023-01-13

**Authors:** Hamid Saeed, Khubaib Ali, Muhammad Nabeel, Muhammad Fawad Rasool, Muhammad Islam, Furqan Khurshid Hashmi, Amna Saeed, Zikria Saleem

**Affiliations:** 1Department of Pharmaceutics, College of Pharmacy, University of the Punjab, Allama Iqbal Campus, Lahore 54000, Pakistan; 2Akhtar Saeed College of Pharmaceutical Sciences, Bahria Town, Lahore 54000, Pakistan; 3Cancer Care Hospital & Research Centre, Lahore 54000, Pakistan; 4Department of Pharmacy Practice, Faculty of Pharmacy, Bahauddin Zakariya University, Multan 60800, Pakistan; 5Department of Pharmacy Administration and Clinical Pharmacy, School of Pharmacy, Xi’an Jiaotong University, Xi’an 710049, China

**Keywords:** COVID-19, vaccine, pharmacy, non-pharmacy, hesitancy, barriers, knowledge, attitudes, Pakistan

## Abstract

University students are a sub-group of the population at high risk of COVID-19 infection, and their judgments on vaccination affect the public attitudes towards vaccination. Thus, the present study aimed to assess the knowledge, attitudes, perceptions, and acceptance of the COVID-19 vaccine among pharmacy and non-pharmacy students. A cross-sectional study was conducted by enrolling pharmacy (375) and non-pharmacy (225) students from the universities in Lahore. Chi-square analysis was used for significant frequency distributions and a 5-point Likert scale was used to score attitude, perception, and acceptance. The majority of the students were aged between 19–24 years, hailing from urban and middle-class families with good self-reported health. The preferred vaccine was Pfizer, followed by Sinopharm and Sinovac. The major source of information was social media, followed by government campaigns and family members. The pharmacy students demonstrated better knowledge and positive attitudes toward COVID-19 vaccination. The non-pharmacy students scored higher for the questions based on scientific leads, myths, and baffling conspiracies. The non-pharmacy students showed higher hesitancy/barrier total scores related to their trust in the health system, COVID-19 vaccine storage, and efficacy. Data suggested that pharmacy students exhibited better knowledge, positive attitudes, and perceptions about COVID-19 vaccination. Overall, vaccine efficacy and safety were mutual concerns. Nonetheless, non-pharmacy students were hesitant due to mistrust in the health system of Pakistan.

## 1. Introduction

A coronavirus-related pneumonia outbreak was discovered in Wuhan, China, in December 2019. Since then, the outbreak expanded rapidly over the world, with tens of thousands of cases in more than 160 nations [1]. At the Munich Security Conference in 2020, Tedros Adhanom Ghebreyesus, WHO’s director-general said, “We’re not only currently fighting a pandemic; we’re fighting an infodemic”. Since the beginning of COVID-19, the prevalence of misleading information, fake news, and conspiracy theories have greatly increased [2]. Conflicting viewpoints and detrimental beliefs are frequently sparked by discussions among people. In particular, the development of vaccines to prevent COVID-19 is the most frequently discussed subject. The flash flood of materials from many sources about COVID-19 and vaccines, most of it contradictory, has already generated a lot of debate and, in some cases, has been branded as “misinformation” [3].

The knowledge, attitude, and perception about an illness, particularly an infectious illness, depends mainly on the severity, spread, and case fatality rates [4]. Though the knowledge, attitudes, and perception about COVID-19 vaccination improved over time, several misconstrued and deluded pieces of information are still being shared and communicated on social media platforms [4]. The attitudes, perceptions, and behaviors of patients toward lifestyle changes are influenced by the health-related behavior of health professionals, since one of the most compelling predictors of a patient’s behavior towards preventive and healthy behavior is the health professional’s own behavior toward such practices [5]. The researcher from the USA showed that the overall vaccine hesitancy among students and trainees in healthcare was 18.9% worldwide—a rate similar to practicing healthcare providers [6]. One out of every 10 Indian medical students was hesitant about COVID-19 vaccination due to concerns regarding the safety and efficacy of the vaccine [7]. Similarly, only 1.26% of Pakistani medical students believed that the COVID-19 vaccine may not be as effective as other well-tested vaccines [8]. According to studies, a strong association exists between health professionals’ knowledge and attitudes toward vaccination and the recommendations given to their patients [5]. 

Throughout the COVID-19 pandemic, pharmacists have been frontline workers in both in- and outpatient settings for the prevention and management of COVID-19 [9], while legislation has been extended to even allow pharmacy interns to administer the COVID-19 vaccine to adults and children [10]. Thus, students in the healthcare discipline, particularly medicine and pharmacy, although not limited to these, may play a pivotal role in the conflict resolution of vaccine hesitancy and anti-vaccination drive. In this context, the pharmacy students, as future healthcare workers, having adequate knowledge about the COVID-19 vaccine, may be inured with the responsibility to provide vaccine recommendations and counseling to vaccine hesitant patients [11]. 

Numerous reports from the Czech Republic, Italy, Lebanon, and Bangladesh suggested that socio-demographic factors, behavioral patterns, illiteracy, lack of trust, availability, and the side effects of the COVID-19 vaccine contribute towards vaccine hesitancy [12,13]. After healthcare workers, Pakistan’s government has mandated to vaccinate university students on a priority basis. Thus, it is pivotal to assess student’s knowledge, attitudes, perceptions, and acceptance towards the COVID-19 vaccine to not only improve their potential in educating the public against anti-vaccine drives but also to determine knowledge- and acceptance-related gaps between university students of various disciplines regarding COVID-19 vaccine, in this case pharmacy vs. non-pharmacy students. Thus, the present study aimed to assess the knowledge, attitudes, perception, and acceptance of the COVID-19 vaccine among pharmacy and non-pharmacy students of Lahore, Pakistan. 

## 2. Materials and Methods

### 2.1. Ethical Approval

The Human Ethical Committee University College of Pharmacy, University of the Punjab, Lahore, approved the study, reference number PUCP/1134/2022. Informed verbal consent was obtained from all the participants while distributing questionnaires.

### 2.2. Study Design

Descriptive, cross-sectional research was undertaken with respondents from Pakistan’s oldest and largest institutions, including the University of the Punjab and numerous other universities in Lahore. The study duration was six months, from February 2022 to June 2022. Data was obtained from both pharmacy and non-pharmacy departments (fine arts, language, social and Islamic studies, social and administrative sciences, and pure sciences and engineering) of the universities after approvals from respective heads. Thereafter, data were segregated into pharmacy and non-pharmacy students for comparative analysis.

### 2.3. Study Population

A total of 600 students, pharmacy (*n* = 375) and non-pharmacy (*n* = 225), from the University of Punjab and other universities constituted our test sample. A sampling frame consisted of undergraduate and master’s students enrolled at the universities. Rao-soft software estimated a sample size of 584 with a 5% margin of error and a 95% confidence interval, with a 50% expected response with expected population size of students ~2 million attending universities of Lahore. However, almost 800 students were approached for data collection. The research participants were enrolled using following inclusion and exclusion criteria:

*Inclusion criteria:* All the students, enrolled in undergraduate or master’s degree programs at the university either in pharmacy or non-pharmacy subjects (fine arts, language, social and Islamic studies, social and administrative sciences, pure sciences and engineering), irrespective of age, gender, ethnicity, religion, social class and willing to participate were enrolled in the study. 

*Exclusion criteria:* All post-graduate students and those not willing to participate were excluded. 

### 2.4. Data Collection 

Data were collected using a comprehensive instrument of a measure, designed after an extensive literature review [7,8,14,15,16,17].

The questionnaire was sent to subject experts/academicians for content validation, and thereafter their expert opinion was incorporated to make the questionnaire simpler and more objective-driven. The reliability of the questionnaire was evaluated with Cronbach’s alpha (0.78), with SPSS version 22. Face validation of the questionnaire was done by conducting a pilot study on 20 students after feedback from the students during the pilot study. The data obtained during the pilot study was not included in the final analysis.

The self-administered questionnaire was distributed to selected students, who were asked to submit the filled forms at the department admission office for final collection. A total of 800 questionnaires were distributed, out of which only 600 questionnaires were finalized for inclusion—incomplete forms (*n* = 133) and those not returned (*n* = 67) to the admission office were considered dropouts. The questionnaire was divided into the following 5 sections. 

*Section 1*: basic demographic, such as age, gender, marital status, area of residence, socioeconomic status, the field of study, and overall health condition.

*Self*-reported health was estimated by asking a single question: ‘How would you rate your health?’, and by providing them response options; good—no health issues (mental or physical), fair—mild health issues (mental or physical), poor—major health issue (mental or physical) that must be treated.

*Section 2*: Fifteen statements were included regarding knowledge about the COVID-19 vaccine and vaccines in general to assess the relevant information and the response to the question was taken on a correct/incorrect basis.

*Section 3*: Attitudes and perceptions regarding the COVID-19 vaccine were assessed utilizing twenty-one statements based upon a 5- point Likert scale from 1 = strongly disagree to 5 = strongly agree.

*Section 4*: Hesitancy and barriers regarding the COVID-19 vaccine were assessed utilizing nineteen statements based upon a 5- point Likert scale from 1 = strongly disagree to 5 = strongly agree.

*Section 5**:* sources of vaccine information and COVID-19 vaccine preferences such as Pfizer (USA), AstraZeneca (UK), Sputnik (Russia), Sinovac (China), Sinopharm (China) and Moderna (USA).

### 2.5. Data Analysis

The data were analyzed using SPSS (IBM, version 22) unless otherwise reported. Descriptive analysis was performed to estimate the percentages and frequencies of the two arms. Associations of dependent variables including knowledge, attitude, beliefs and self-perceived effectiveness and independent variables, such as demographics, were estimated using Pearson’s Chi-square. An alpha value of 0.05 or less was considered statistically significant.

## 3. Results

### 3.1. Demographics of the Students, Pharmacy (P) and Non-Pharmacy (NP)

Demographics of the students, pharmacy (**P**) and non-pharmacy (**NP**), are summarized in Table 1. Data revealed that the majority of the students were between 19–24 years of age (**P:** 81.3%, **NP:** 72.9%, *p* = 0.02). More than 50% of the students were females (**P:** 68%, **NP:** 50.2%, *p* = 0.001), mostly unmarried (*p* = 0.07), hailing from urban areas (*p* = 0.118), belonging to middle-class families (**P:** 91.5%, **NP:** 91.6%, *p* = 0.473), with the majority of the student’s self-reported health status being good (**P:** 63.7%, **NP:** 60%, *p* = 0.633) (Table 1). 

### 3.2. Sources of Information and Vaccine Preferences among Pharmacy and Non-Pharmacy Students

Notable sources of information and preferences of pharmacy and non-pharmacy students regarding COVID-19 vaccination are summarized in Table 2. Social media (**P:** 44.5%, **NP:** 37.3%, *p* = 0.019) and government advertisements were found as the major sources of information about COVID-19 vaccines. The pfizer vaccine was majorly preferred by pharmacy students (**P:** 48%, **NP:** 32%, *p* = 0.005), while Sinovac (**P:** 20.3%, **NP:** 23.1%, *p* = 0.005) and Sinopharm (**P:** 20%, **NP:** 27.6%, *p* = 0.005) were majorly preferred by non-pharmacy students (Table 2). 

### 3.3. Assessment of Student’s Knowledge, P and NP, about COVID-19 Vaccine Information

Knowledge of students about COVID-19 vaccines, pharmacy (**P**), and non-pharmacy (**NP**) are summarized in Table 3. The pharmacy (**P**) students demonstrated better knowledge about the COVID-19 vaccination than non-pharmacy (**NP)** students. Data revealed that the majority of the students knew about the viral nature of COVID-19 (**P:** 99.5%, **NP:** 96.9%, *p* = 0.012). More than 90% of the pharmacy students knew that the COVID-19 vaccine can reduce the spread of disease (**P:** 90.1%, **NP:** 82.7%, *p* = 0.008) and the severity of disease symptoms (**P:** 90.1%, **NP:** 88.4%, *p* = 0.514), being well acquainted with the process of getting vaccinated (**P:** 90.7%, **NP:** 87.6%, *p* = 0.229). Additionally, the majority of the pharmacy students understood the need for more than one dose(s) of vaccine (**P:** 94.1%, **NP:** 86.7%, *p* = 0.002) and that the vaccine stimulates the immune system to produce antibodies (**P:** 90.7%, **NP:** 80%). The majority of the pharmacy students believed that, after vaccination, there was a need to follow preventive measures (**P:** 88%, **NP:** 80.9%, *p* = 0.017) and that antibiotics were not an alternative to the COVID-19 vaccine (**P:** 76.8%, **NP:** 64.4%, *p* = 0.001) (Table 3).

### 3.4. Assessment of Attitudes and Perception of Students, Pharmacy vs. Non-Pharmacy

Respondent’s attitudes and perceptions were scored using a 5-point Likert scale, as summarized in Table 4. Data revealed that NP students scored higher for the questions related to misconceptions and myths about the COVID-19 vaccine, such as the use of fetal tissues in vaccine development (**P:** 2.74 ± 1.04, **NP:** 3.04 ± 0.098, *p* = 0.00046), with a highest effect size of 0.297, that it contains controversial substances (**P:** 2.74 ± 1.09, **NP:** 2.98 ± 1.08, *p* = 0.011), causes infertility (**P:** 2.33 ± 1.10, **NP:** 2.63 ± 1.18, *p* = 0.002), with an effect size of 0.263, and contains 5G nano-chips to control people (**P:** 2.25 ± 1.13, **NP:** 2.48 ± 1.29, *p* = 0.023) with an effect size of 0.190. Conversely, pharmacy students scored higher for the question about the normalization of life after COVID-19 vaccination (**P:** 3.17 ± 1.07, **NP:** 2.94 ± 1.19, *p* = 0.018). No significant differences were found in the observed scores for the rest of the items regarding attitudes and perceptions (Table 4).

### 3.5. Assessment of COVID-19 Vaccine Hesitancy and Barriers among Pharmacy and Non-Pharmacy Students

The COVID-19 vaccine acceptance was assessed using a 5-point Likert scale between pharmacy and non-pharmacy students summarized in Table 5. Data suggested that compared to pharmacy students, non-pharmacy students scored higher for the items questioning Pakistan’s healthcare system, such as lack of trust in Pakistan’s healthcare system (**P:** 3.18 ± 1.28, **NP:** 3.46 ± 1.21, *p* = 0.011) with size effect of 0.225 and improper storage of vaccines (**P:** 2.94 ± 1.21, **NP:** 3.19 ± 1.07, *p* = 0.013). Additionally, positive scores were obtained for NP students for the items that signified their fear of injections/needles (**P:** 2.46 ± 1.27, **NP:** 2.78 ± 1.35, *p* = 0.004) with effect size of 0.244 and about contracting COVID-19 at the vaccination centers (**P:** 2.68 ± 1.13, **NP:** 2.92 ± 1.10, *p* = 0.012) having effect size of 0.215. Though surprisingly, NP students demonstrated higher scores when asked that COVID-19 vaccines will not stop the spread of COVID-19 (**P:** 2.55 ± 1.08, **NP:** 2.77 ± 1.14, *p* = 0.020) with effect size of 0.2 (Table 5). 

## 4. Discussion

Numerous studies have shown that students representing health education sectors demonstrated vaccine hesitancy, e.g., 46% in Egypt [18], 23% in the USA [19], 10.6% in India [7], 17.8% in Israel [20], 62.7% in Uganda [21] and 75.5% in Zambia [22]. Thus, the present study evaluated knowledge, attitudes, perceptions, and acceptance of the COVID-19 vaccine among pharmacy and non-pharmacy students of Lahore, Pakistan. Data revealed that the majority of the students were between 19–24 years of age, hailing from urban areas with good self-reported health. The students majorly rely on social media information, yet pharmacy students preferred Pfizer, and NP students preferred Chinese vaccines, Sinovac and Sinopharm, for vaccination. Overall, pharmacy students exhibited better knowledge, positive attitudes, and perceptions toward COVID-19 vaccination, yet efficacy and safety were major concerns for all the students. The NP students still believed in certain myths due to misconceptions. In this context, non-pharmacy students scored higher on a 5-point Likert scale for a few hesitancy and barrier questions, demonstrating vaccine acceptance challenges. 

A fundamental challenge for the health authorities is the acceptance of the COVID-19 vaccine among healthcare workers and the public across countries worldwide. Poor health literacy, lack of knowledge, delusional beliefs, and conspiracy theories about the COVID-19 vaccine pose a serious challenge to vaccination in Pakistan [23,24]. With overall good self-reported health, social media was the most relied upon source of information to both pharmacy (44.5%) and non-pharmacy (37.3%) students, followed by the information provided by the government authorities (24.4–26.4%). These findings, that social media, and the internet were the major sources of information for students, were in complete agreement with previous studies from Bangladesh (33.4%), Vietnam (87.61%), Jordan (83.4%), and Egypt (75.5%) [25]. However, as acknowledged in previous studies [26,27], some credit goes to the government of Pakistan, as government authorities were rated as the second major source of information by the students. However, it is pertinent to mention that extreme care should be exercised while relying on social media and social network information, since our previous findings indicated that almost 58% of sources of myths and misconceptions about the COVID-19 vaccine came from social media/networks [28]. Furthermore, compared to NP students, P students demonstrated better knowledge in terms of the causative agent of the disease, the role of the vaccine in reducing the spread of the disease, dosing requirements, and its impact on the immune system. The knowledge-based misconceptions were higher among NP students, e.g., the dose requirements, impact of vaccines on the spread and even cure of the disease, and the chances of auto-immune disease with vaccination. Similar to these findings, a study from Pakistan demonstrated better knowledge among those acquiring medical education [26], a study from Bangladesh showed positive knowledge among 58.13% of university students [29], and 42.4% of university students in the UAE demonstrated good knowledge about COVID-19 vaccine [4]. Nevertheless, for the items with incorrect answers, NP students, >84%, answered otherwise, e.g., antibiotics can be alternatives to vaccines, no need to follow personal protective measures, and the post-vaccination chances of auto-immune disease. Yet, a certain number of pharmacy students also responded incorrectly to knowledge items, possibly due to the inability of the students to cross-verify information when social media information is the major source, corroborating several previous reports [4,29,30].

It has been well documented that meaningful knowledge about the disease and therapy would impact the patient’s attitudes and perceptions toward future disease development and early prevention [24,31,32]. Overall, comparable mean scores were obtained between pharmacy and non-pharmacy students for the attitudes and perceptions questions, namely, that efficacy and safety cannot be trusted, that the COVID-19-infected person does not need a vaccine, that current vaccines do not protect against COVID-19 variants, that vaccines might change ones DNA, and that vaccination is a business strategy by big pharma companies, etc. Additionally, NP students’ mean scores were higher in comparison to pharmacy students for the questions related to vaccine fallacy—vaccines were made from fetal tissues with effect size of 0.297, depicting moderate effect size of non-pharmacy education on this particular item, vaccines cause infertility (effect size 0.263), having moderate effect size on this item in NP students, and vaccines contain 5G nano-chip to contain and trace people. Such fallacies and fakeries have been reported previously among university students and the public regarding COVID-19 vaccines [33,34]. These data and above the knowledge responses suggested that both pharmacy and non-pharmacy students require specific education and training, not only to improve informed decisions towards COVID-19 vaccination but also to prepare them, particularly pharmacy students, to spread awareness and give recommendations or counsel to the vaccine-hesitant individuals. However, NP students’ mean scores were higher for the responses related to barriers and hesitancy, namely, lack of trust in Pakistan’s health system, with moderate effect size of 0.225, improper vaccine storage, fear of injection (effect size of 0.244), fear of contracting COVID-19 infection at the vaccination center, and belief that vaccines will not prevent COVID-19 infection. Overall, P and NP students showed low levels of mean hesitancy scores for several of the hesitancy questions. However, the high-grade hesitancy scores regarding efficacy and safety, the emphasis on the post-vaccine surveillance to assess the efficacy and safety concerns, and the lack of trust in government were found to be the major hesitancy/barriers among the students—posing vaccine acceptance challenges. 

The data from our study and previous reports vividly state that challenges to vaccine hesitancy in university students are not new. However, they are diverse, with beliefs such as that there is insufficient information about efficacy and safety, lack of trust in vaccine sources and the government health system, and a concern over unforeseeable future vaccine-related health implications. Thus, overall, with more than above average knowledge, positive attitudes and perceptions, a few myths and misconceptions, and low-grade hesitancy among students— the future educators, recommenders and counselors of anti-vaxxers—offering sufficiently effectual information, modification in the information based on socio-cultural needs, targeted interventions to dispel the myths, and hitting upon and regulating the sources of information conferring false perception about COVID-19 vaccines can help increase vaccine acceptance among university students. 

The present study has several limitations, which could be addressed in the prospective research proposals. The cross-sectional design of the study does not allow us to document the responses over time—which are subject to change over time. The self-reported responses might be subject to bias due to an individual’s recall ability, honesty, utmost surroundings, and psychological state. The study is limited to the pharmacy and non-pharmacy students of Lahore, and thus the results cannot be generalized to students of other health professions and universities of other provinces. 

Thus, any attempt to address these limitations might accentuate the future research on this topic. 

## 5. Conclusions

The study concluded that the majority of the students, pharmacy and non-pharmacy, resort to social media as a major source of information, followed by the information provided by the government authorities. The preferred vaccine choice was Pfizer by pharmacy students, while Chinese vaccines, Sinovac and Sinopharm by non-pharmacy students. Pharmacy students showed better knowledge of the items based on scientific leads, positive attitudes, and perceptions in terms of the normalization of life, which might be attributable to study area. Nevertheless, both, P and NP students were concerned about the efficacy and safety of vaccines. Majorly, only NP students believed in vaccine-related myths and delusional misconceptions about the COVID-19 vaccines, such as using fetal tissues and causing infertility. Hesitancy scores were higher in NP students for the items questioning the trust level in Pakistan’s health system and the efficacy and safety of the COVID-19 vaccines. 

## Figures and Tables

**Table 1 vaccines-11-00176-t001:** Demographics of the students, pharmacy and non-pharmacy.

Characteristics	Pharmacy Students, *n* = 375 (*%*)	Non-Pharmacy Students, *n* = 225 (*%*)	*p* Values
**Age** *≤18* *19–24* *≥25*	60 (16) 305 (81.3) 10 (2.7)	41 (18.2) 164 (72.9) 20 (8.9)	**0.02 ***
**Gender** *Male* *Female*	120 (32) 225 (68)	112 (49.8) 113 (50.2)	**0.001 ****
**Marital Status** *Married* *Unmarried*	23 (6.1) 352 (93.9)	28 (12.4) 197 (87.6)	0.07
**Area of Residence** *Urban* *Rural*	305 (81.3) 70 (18.7)	171 (76.0) 54 (24.0)	0.118
**Socioeconomic Status** *Lower* *Middle* *Upper*	12 (3.2) 343 (91.5) 20 (5.3)	4 (1.8) 206 (91.6) 15 (6.7)	0.473
**Overall, Health** *Good* *Fair* *Poor*	239 (63.7) 124 (33.1) 12 (3.2)	135 (60.0) 83 (36.9) 7 (3.1)	0.633

*p* values: * 0.05–0.002, ** < 0.002.

**Table 2 vaccines-11-00176-t002:** Sources of Information and COVID-19 Vaccine Preferences among Pharmacy and Non-pharmacy students.

Parameters	Pharmacy Students, *n* = 375 (*%*)	Non-Pharmacy Students, *n* = 225 (*%*)	*p* Values
**Sources of Information** *Social media* *News channels* *Govt. Authorities* *Family/friends*	167 (44.5) 40 (10.7) 99 (26.4) 69 (18.4)	84 (37.3) 44 (19.4) 55 (24.4) 42 (18.7)	**0.019 ***
**Vaccine Preferences** *Pfizer* *AstraZeneca* *Sputnik* *Sinovac* *Sinopharm* *Moderna*	180 (48) 11 (2.9) 7 (1.9) 76 (20.3) 75 (20) 26 (6.9)	72 (32) 12 (5.3) 4 (1.8) 52 (23.1) 62 (27.6) 23 (10.2)	**0.005 ***

*p* values: * 0.05–0.002.

**Table 3 vaccines-11-00176-t003:** Assessment of COVID-19 Vaccine’s Knowledge among Students, Pharmacy and Non-Pharmacy.

Questions	Pharmacy Students, *n* = 375 (*%*)	Non-Pharmacy Students, *n* = 225 (*%*)	*p* Value
*COVID-19 is a Viral disease.* **Correct** **Incorrect**	373 (99.5) 2 (0.5)	218 (96.9) 7 (3.1)	**0.012 ***
*COVID-19 Vaccine can reduce the spread of disease in the community* **Correct** **Incorrect**	338 (90.1) 37 (9.9)	186 (82.7) 39 (17.3)	**0.008 ***
*COVID-19 Vaccine can help reduce the severity of COVID-19 disease symptoms* **Correct** **Incorrect**	338 (90.1) 37 (9.9)	199 (88.4) 26 (11.6)	0.514
*COVID-19 Vaccine can cure COVID-19 Disease* **Correct** **Incorrect**	174 (46.4) 201 (53.6)	106 (47.1) 119 (52.9)	0.866
*Antibiotics are alternative to the COVID-19 Vaccine* **Correct** **Incorrect**	87 (23.2) 288 (76.8)	80 (35.6) 145 (64.4)	**0.001 ****
*Everyone including children can receive the COVID-19 Vaccine* **Correct** **Incorrect**	269 (71.7) 106 (28.3)	161 (71.6) 64 (28.4)	0.963
*After getting the COVID-19 Vaccine I don’t need to follow preventive measures* **Correct** **Incorrect**	45 (12) 330 (88)	43 (19.1) 182 (80.9)	**0.017 ***
*Vaccination process in Pakistan is simple—register* via *SMS from cell phone* **Correct** **Incorrect**	340 (90.7) 35 (9.3)	197 (87.6) 28 (12.4)	0.229
*COVID-19 vaccine can also protect from Influenza* **Correct** **Incorrect**	119 (31.7) 256 (68.3)	64 (28.4) 161 (71.6)	0.397
*There is more than one dose(s) of the Vaccine* **Correct** **Incorrect**	353 (94.1) 22 (5.9)	195 (86.7) 30 (13.3)	**0.002 ***
*Vaccination increases the chances of auto-immune diseases* **Correct** **Incorrect**	127 (33.9) 248 (66.1)	104 (46.2) 121 (53.8)	**0.003 ***
*There is only ONE type of COVID-19 vaccine available in Pakistan* **Correct** **Incorrect**	40 (10.7) 335 (89.3)	36 (16) 189 (84)	0.057
*COVID-19 vaccine stimulates our immune system to produce antibodies against the COVID-19 virus* **Correct** **Incorrect**	340 (90.7) 35 (9.3)	180 (80) 45 (20)	**0.0001 ****

*p* values: * 0.05–0.002, ** < 0.002.

**Table 4 vaccines-11-00176-t004:** Assessment of Attitude and Perception of Students towards COVID-19 Vaccination.

Attitudes & Perception Questions	Pharmacy Students, *n* = 375 (*%*)	Non-Pharmacy Students, *n* = 225 (*%*)	*p* Values
*If I already had COVID-19 infection I don’t need a vaccine*	2.47 ± 1.28	2.51 ± 1.41	0.747
*Due to its emergent approval, its efficacy and safety cannot be trusted*	3.09 ± 1.05	3.06 ± 1.07	0.707
*One might get COVID-19 disease due to the COVID-19 vaccine*	2.73 ± 1.19	2.80 ± 1.25	0.523
*COVID-19 can cause serious side effects*	3.13 ± 1.15	3.30 ± 1.16	0.080
*Current COVID-19 vaccines do not protect against new COVID variants*	3.15 ± 1.06	3.32 ± 1.08	0.067
*Children didn’t get severe disease symptoms, so they don’t need the COVID-19 vaccine*	2.54 ± 1.04	2.50 ± 1.29	0.706
*COVID-19 vaccines were made using fetal tissues*	2.74 ± 1.04	3.04 ± 0.98	**0.00046 ****
*One does not need two doses of COVID-19 vaccines*	2.31 ± 1.12	2.32 ± 1.21	0.863
*COVID-19 Vaccine was developed with or contains controversial substances*	2.74 ± 1.09	2.98 ± 1.08	**0.011 ***
*Life will get normal after vaccination*	3.17 ± 1.07	2.94 ± 1.19	**0.018 ***
*Only old or people with underlying diseases should get the COVID-19 Vaccine*	2.25 ± 1.12	2.33 ± 1.15	0.437
*COVID-19 Vaccine causes Infertility*	2.33 ± 1.10	2.63 ± 1.18	**0.002 ***
*COVID-19 Vaccine contains 5G Nano chips to contain people*	2.25 ± 1.13	2.48 ± 1.29	**0.023 ***
*Proper protocol wasn’t followed for the development of the COVID-19 vaccine*	2.83 ± 1.15	3.01 ± 1.12	0.059
*COVID-19 Vaccine is a business strategy for Big Pharmaceutical Companies*	2.89 ± 1.24	2.85 ± 1.16	0.678
*COVID-19 Vaccine changes your DNA*	2.27 ± 1.13	2.4 ± 1.22	0.196
*Vaccines should be administered free of charge*	4.128 ± 0.99	4.11 ± 1.00	0.882
*COVID-19 vaccine can also protect against influenza infection*	3.00 ± 1.07	3.10 ± 2.28	0.773

*p* values: * 0.05–0.002, ** < 0.002. Responses were based on a 5-point Likert-type scale with 1 = Strongly Disagree 2 = Disagree 3 = Neutral 4 = Agree 5 = Strongly Agree. Mean Attitude scores greater than 3 (midpoint/neutral score) were treated as high, and those below 3 as having a low mean score.

**Table 5 vaccines-11-00176-t005:** Assessment of COVID-19 Vaccine Acceptance among Students; pharmacy and non-pharmacy.

Acceptance Questions	Pharmacy Students, *n* = 375 (*%*)	Non-Pharmacy Students, *n* = 225 (*%*)	*p* Values
*I don’t need a vaccine as I am Healthy and Young*	2.29 ± 1.27	2.35 ± 1.31	0.568
*Government promote vaccination for political & financial gains*	2.80 ± 1.20	2.87 ± 1.21	0.476
*I believe that COVID-19 is not a real disease*	2.02 ± 1.07	2.14 ± 1.13	0.196
*I believe that vaccination affects natural immunity*	2.83 ± 1.20	2.94 ± 1.10	0.289
*I do not trust Pakistan’s health system*	3.18 ± 1.28	3.46 ± 1.21	**0.011 ***
*Death is inevitable and the vaccine cannot prevent it*	3.17 ± 1.35	3.33 ± 1.30	0.139
*My religion does not permit me to get vaccine*	2.12 ± 1.13	2.28 ± 1.22	0.108
*Lack of access to vaccination centers*	2.63 ± 1.11	2.76 ± 1.26	0.169
*Vaccine storage is not proper in Pakistan*	2.94 ± 1.21	3.19 ± 1.07	**0.013 ***
*Vaccines in Pakistan are donations that’s why not effective and safe*	2.61 ± 1.10	2.74 ± 1.10	0.177
*It is against my Morals and Beliefs to get Vaccinated*	2.09 ± 1.05	2.11 ± 1.10	0.822
*I feel insecure about the Vaccines offered by the Health Authorities*	2.58 ± 1.17	2.68 ± 1.22	0.332
*I am afraid of needles/Injections*	2.46 ± 1.27	2.78 ± 1.35	**0.004 ***
*I am afraid that I will contract COVID-19 at the vaccination center*	2.68 ± 1.13	2.92 ± 1.10	**0.012 ***
*I am concerned about the efficacy and side effects caused by the COVID-19 Vaccine*	3.14 ± 1.14	3.08 ± 1.05	0.558
*I don’t need the vaccine because I follow preventive measures*	2.29 ± 1.09	2.40 ± 1.16	0.245
*I don’t believe that the vaccine will stop the COVID-19 disease*	2.55 ± 1.08	2.77 ± 1.12	**0.020 ***
*I am waiting for more people to get vaccinated*	3.10 ± 1.31	3.05 ± 1.34	0.634

*p* values: * 0.05–0.002. Responses were based on a 5-point Likert-type scale with 1 = Strongly Disagree 2 = Disagree 3 = Neutral 4 = Agree 5 = Strongly Agree. Mean Attitude scores greater than 3 (midpoint/neutral score) were treated as high, and those below 3 as having a low mean score.

## Data Availability

Data will be available on reasonable request to the corresponding author.

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
