# Peer review of "Knowledge, Attitudes, Perceptions, and Acceptance of COVID-19 Vaccination among Pharmacy and Non-Pharmacy Students"

_vaccines, 2023, doi:10.3390/vaccines11010176_

Round 1
Reviewer 1 Report
The paper is well written.
I have mainly 4 questions.
1. Characteristics of NP and P students are different. We will not be able to see if these characteristics are driving the results, or it is the study (NP vs P) area that students study at university that is driving the results (in terms of perception, knowledge etc of vaccine). Can you address this issue?
2. Why P students have higher knowledge / positive perceptions? Any evidence that you can show?
3. Why focusing only P vs. NP? What about medical students, science students etc?
4. I am still not sure about the policy implications. Can you clarify?
Author Response
Reviewer #1:
- Characteristics of NP and P students are different. We will not be able to see if these characteristics are driving the results, or it is the study (NP vs P) area that students study at university that is driving the results (in terms of perception, knowledge etc of vaccine). Can you address this issue?
Response: We thank the reviewer for the valuable comments. As far as characteristics are concerned, demogrpahics, mainly age and gender frequencies were significantly different, the rest were insignificantly different. Therefore, based on these findings, we believe, and as riglty pointed out by you, it’s the study area that could be the main reason behind the observed differences. Thank You.
Why P students have higher knowledge / positive perceptions? Any evidence that you can show?
Response: Number of national and international studies reported that students acquiring medical or pharmacy education had better knowledge and positive perception and also referenced in our discussion, e.g.,
Serbezova, A.; Mangelov, M.; Zaykova, K.; Nikolova, S.; Zhelyazkova, D.; Balgarinova, N.J.B.; Equipment, B. Knowledge and attitude toward COVID-19 vaccines amongst medical, dental and pharmacy students. A cross-sectional study from Bulgaria. Int J Environ Res Public Health 2021, 35, 2046-2054.
Usman, J.; Ifrah Arshad, A.F.; Ahsan, M.; Mina, N.J.J.o.R.M.C. Knowledge and attitude pertinent to COVID-19 and willingness to COVID vaccination among medical students of University College of Medicine & Dentistry Lahore. J Rawalpindi Med Coll 2021, 25.
Rosental, H.; Shmueli, L.J.V. Integrating health behavior theories to predict COVID-19 vaccine acceptance: differences between medical students and nursing students. Vaccines 2021, 9, 783.
Comparison of medical and non-medical students (https://www.ncbi.nlm.nih.gov/pmc/articles/PMC9359383/) – medical students have improved knowledge and acceptance.
However, a direct comparison, as reported in our study, has not been reported before, therefore, a similar evidence is still difficult to furnish as reported in our findings. Nonetheless, we tried to make this point here that compared to non-pharmacy students the improved knowledge and positive perception in P students must be associated with their educational background related to medical or pharmaceutical sciences field, because with minimal differences in the demographic characteristics the observed differences could be attributed to field of study. Additonally, if we look at the information sources, the trend appeared similar in both groups, e.g., major sources social media followed by government authorities.Thank You.
- Why focusing only P vs. NP? What about medical students, science students etc?
Response: As mentioned in the method section we excluded particularly biological sciences students due to similarities in some of the subjects offered to pharmacy and biological sciences students. However, already number of studies on medical and non-medical students have been reported, yet not a single on pharmacy and non-pharmacy students. Therefore, keeping in view this objective in mind we choose to select this population for our study. Thank You.
- I am still not sure about the policy implications. Can you clarify?
.
Response: We tried to emphasize on the socio-cultural differences, deluge of information on social media – mostly unreferenced, and those related to study area, thus, the targeted interventions by the policy makers and health officials must be devised and planned taking into account these differences – to target misconceptions/myths among non-pharmacy students and providing correct and effectual information to pharmacy students to dispel the false perceptions. Thank You.

Reviewer 2 Report
The study is simple well designed and written
Minor language edits are needed
Author Response
January 07, 2022
Editor in Chief
Vaccines
We are thankful for a meticulous and critical review, aimed at improving our manuscript. Please consider the revised manuscript with submission ID: Vaccines.2152889.R1- titled: “Knowledge, Attitudes, Perceptions, and Acceptance of COVID-19 Vaccination among Pharmacy and Non-Pharmacy Students” by Saeed et al. for publication in the Journal.
We have now revised the manuscript and made changes according to the editor’s comments. At this stage, we are providing the manuscript with TRACK CHANGES. Detailed response is also given below.
We really hope now that the revised version of the manuscript will be acceptable for publication in the Journal.
With my best regards,
Hamid Saeed, PhD
Professor
University College of Pharmacy
University of the Punjab
Tel. +92304-880-1243
e-mal. hamid.pharmacy@pu.edu.pk
Response to editor/reviewers
Point-by-point rebuttal to editor’s and editor/reviewers comments
Reviewer #2:
Minor language edits are needed
Response: We thank the reviewer for the valuable comments and meticulous review of our manuscript aimed at improving the manuscript. We have critically reviwed the manuscript for language mistakes and will try to improve the conclusion. Thank You.

Reviewer 3 Report
Dear authors,
I have now completed the review of the manuscript titled "Knowledge, Attitudes, Perceptions, and Acceptance of COVID-19 Vaccination among Pharmacy and Non-Pharmacy Students."
In the present study, the authors conducted cross-sectional study by enrolling Pharmacy and non-Pharmacy students from the universities of Lahore.
The manuscript is interesting and, in general, fair written.
I have some suggestions to further improve the quality of the manuscript.
1. The background section introduced some relevant articles. Please explain the results or summarize with effect sizes.
2. I wonder the original data is firstly used for this research. If so, kindly consider making it public.
3. What is the overall health good/fair/poor? The expression is vague, please explain readers with examples like: fair is a having minor disease but not severe to be treated.
4. What is the future scope of the proposed research, authors have described the limitations in a good way, I suggest that these can be the future scope of the work.
Author Response
January 07, 2022
Editor in Chief
Vaccines
We are thankful for a meticulous and critical review, aimed at improving our manuscript. Please consider the revised manuscript with submission ID: Vaccines.2152889.R1- titled: “Knowledge, Attitudes, Perceptions, and Acceptance of COVID-19 Vaccination among Pharmacy and Non-Pharmacy Students” by Saeed et al. for publication in the Journal.
We have now revised the manuscript and made changes according to the editor’s comments. At this stage, we are providing the manuscript with TRACK CHANGES. Detailed response is also given below.
We really hope now that the revised version of the manuscript will be acceptable for publication in the Journal.
With my best regards,
Hamid Saeed, PhD
Professor
University College of Pharmacy
University of the Punjab
Tel. +92304-880-1243
e-mal. hamid.pharmacy@pu.edu.pk
Response to editor/reviewers
Point-by-point rebuttal to editor’s and editor/reviewers comments
Reviewer #3:
- The background section introduced some relevant articles. Please explain the results or summarize with effect sizes.
Response: We thank the reviewer for the valuable comments and meticulous review of our manuscript aimed at improving the manuscript. We have now tried to explain the results with effect size. Thank You.
- I wonder the original data is firstly used for this research. If so, kindly consider making it public.
Response: We will do that. Thank You.
What is the overall health good/fair/poor? The expression is vague, please explain readers with examples like: fair is a having minor disease but not severe to be treated.
Response: Related explanation has been added in the method section. Thank You.
- What is the future scope of the proposed research, authors have described the limitations in a good way, I suggest that these can be the future scope of the work.
Response: We have now made some changes in the limitation section to also recommend these points as future scope of work. Thank You.
